# COVID-19 pandemic and risk factor measurement in individuals with cardio-renal-metabolic diseases: A retrospective study in the United Kingdom

Sharmin Shabnam[1], Francesco Zaccardi[1], Tom Yates[2], Nazrul Islam[3], Cameron Razieh[1,4], Yogini V Chudasama[1], Amitava Banerjee[5], Samuel Seidu[1], Kamlesh Khunti[1]*, Clare L Gillies[1]

1 Leicester Real World Evidence Unit, Leicester Diabetes Centre, Leicester, United Kingdom, 2 National Institute for Health Research (NIHR) Leicester Biomedical Research Centre (BRC), Leicester Diabetes Centre, Leicester General Hospital, Leicester, United Kingdom, 3 Primary Care Research Centre, University of Southampton, Southampton, United Kingdom, 4 Health Analysis and Life Events Division, Office for National Statistics, Newport, United Kingdom, 5 Institute of Health Informatics, University College London, London, United Kingdom

* kk22@leicester.ac.uk

## Abstract

### Background

Comprehensive research about changes in risk factor (RF) management of people with chronic conditions during the COVID-19 pandemic is sparse. We aimed to study the impact of the pandemic on RF assessment in people with type 2 diabetes (T2DM), cardio-vascular disease (CVD), and chronic kidney disease (CKD).

### Method

Using UK Clinical Practice Research Datalink GOLD, we identified adults with T2DM, CVD and CKD who were alive and registered two years before (March 2018 – February 2019; March 2019 – February 2020) and one year during (March 2020 – February 2021) the pandemic. We estimated the proportion of people whose RFs (systolic (SBP) and diastolic (DBP) blood pressure, total cholesterol (TC), body mass index, smoking, and HbA1c) were assessed, mean values, and the proportion of controlled at each period for each cohort, overall and by age, sex, ethnicity, and deprivation. Multivariable logistic regression was used to estimate the association of patient characteristics (age, sex, ethnicity, deprivation, and comorbidity) with the outcome of having all RFs assessed during a given period.

### Results

Within the T2DM cohort, 66.4% and 65.2% had assessments of HbA1c in 2018 and 2019, which reduced to 43.4% in 2020. In CVD cohort, 76.9% and 72.6% had their BP measurements (54.5% and 51.3% for TC) in 2018 and 2019 respectively, which declined to 40.6%

**Data availability statement:** This study is based in part on data from the Clinical Practice Research Datalink obtained under licence from the UK Medicines and Healthcare products Regulatory Agency. The data is provided by patients and collected by the NHS as part of their care and support. The interpretation and conclusions contained in this study are those of the author/s alone. Protocol for this study is available on CPRD website: https://www.cprd.com/approved-studies/investigating-disruptions-primary-care-result-covid-19-pandemic-and-introduction. The data controller for CPRD (Department of Health and Social Care) does not allow sharing raw data with third parties. Electronic health records are considered sensitive data in the UK by the Data Protection Act and are not publicly available. Researchers may apply for data access at: https://www.cprd.com/research-applications. Statistical code is available upon request from the first author. Phenotypes used to define the cohort, medical conditions, and ethnicity are available at the GitHub link: https://github.com/shabnam-shbd/risk_factor_control_during_covid19.

**Funding:** This research was part of the Data and Connectivity National Core Study, led by Health Data Research UK in partnership with the Office for National Statistics and funded by UK Research and Innovation (grant ref MC_PC_20058). This work was also supported by The Alan Turing Institute via 'Towards Turing 2.0' EPSRC Grant Funding. The funders had no role in study design, data collection and analysis, decision to publish, or preparation of the manuscript.

**Competing interests:** KK has acted as a consultant, speaker or received grants for investigator-initiated studies for Astra Zeneca, Bayer, Novartis, Novo Nordisk, Sanofi-Aventis, Lilly and Merck Sharp & Dohme, Boehringer Ingelheim, Oramed Pharmaceuticals, Pfizer, Roche, Daiichi-Sankyo and Applied Therapeutics. The authors have declared that no competing interests exist.

(30.7% for TC) in 2020. In CKD cohort, BP assessments declined from 77.9% and 72.3% in 2018 and 2019 respectively to 45.0% in 2020. These findings were consistent across patient demographics. In those with T2DM, SBP and DBP increased (+1.65 mmHg and +1.02 mmHg) in 2020. Elderly people were less likely to have all their RFs assessed in 2020 in all three cohorts compared to previous years.

## Conclusions

Among people with major cardiometabolic conditions, there have been substantial reductions in the assessment and control of several key RFs during the pandemic. These patients will need regular monitoring in future for the prevention of complications. Our findings also highlight the need for resilient healthcare systems to ensure continuity of care and mitigate disparities in high-risk populations.

## Introduction

The Coronavirus Disease 2019 (COVID-19) pandemic in 2020 has exerted a widespread impact on global health, with an estimated 7.1 million total deaths recorded as of January 2025 [1]. In addition to the direct effects caused by COVID-19, the indirect effects have also been substantial, particularly regarding healthcare accessibility during and after national lockdowns [2]. Infection control measures, such as social distancing and travel restrictions implemented during the pandemic have undoubtedly helped mitigate the spread of the virus and prevented more deaths. However, the lockdowns led to sweeping changes to healthcare delivery, access, and utilisation [2]; the impact was particularly pronounced for vulnerable and high-risk individuals who were advised to stay at home. The effect is still being felt in the National Health Service (NHS) in the United Kingdom, as waiting lists have lengthened [3] and primary care visits are, in part, still being delivered virtually rather than face-to-face [4]. Since healthcare in the United Kingdom has not fully reverted to the pre-pandemic state, the lingering effects of such disruptions might still exist and continue to affect vulnerable population groups.

COVID-19, caused by the SARS-CoV-2 virus, has disproportionately impacted individuals with pre-existing health conditions, with evidence identifying several key risk factors associated with severe disease and mortality. These include older age, male sex, socioeconomic deprivation, ethnicity, obesity, and chronic conditions such as diabetes, cardiovascular, and respiratory diseases. These risk factors are particularly concerning in populations with cardio-renal-metabolic conditions, as these individuals often experience impaired immune responses when infected with SARS-CoV-2 [5].

People with long-term conditions (LTCs) require regular contact with primary care which allows them to monitor risk factor (RF) control and disease evolution to make necessary adjustments to the treatment plans [6–8]. Previous research has suggested that even short-term delay in risk factor control is associated with worse outcomes in people with chronic diseases [9,10]. Furthermore, comprehensive research specifically on people with LTCs is sparse. Further knowledge about changes in RF management will help health services identify specific vulnerabilities and implement strategies to ensure continuous care amidst competing demands.

This study aimed to analyse the impact of the COVID-19 pandemic on the assessment of key RFs in individuals with cardio-renal-metabolic LTCs, specifically type 2 diabetes (T2DM), cardiovascular disease (CVD), and chronic kidney disease (CKD). These conditions were chosen as they are highly prevalent (globally about 530 million people have diabetes [11]; in the United Kingdom – approximately 5.6 million people have diabetes, 90% of which are type

2 [12]; 6.4 million have CVD [13]; and 3.5 million have CKD stages 3-5 [14]) and they are some of the leading causes of death and disability worldwide. Using routinely collected health-care data, we sought to identify changes in RF monitoring during the pandemic and evaluate differences by demographic and clinical characteristics to inform strategies for improving chronic disease management in future public health emergencies.

## Methods

### Data source and study population

This observational, retrospective cohort study utilising the Clinical Practice Research Data-link (CPRD) GOLD has been conducted and reported in line with the RECORD guidelines (RECORD checklist reported in the **Supplement**). CPRD is a comprehensive research database containing anonymised electronic health records from primary care practices in the UK [15]. The CPRD GOLD database covers approximately 4.6% of the UK population and is considered broadly representative of the UK population in terms of age, sex, and ethnicity [15,16]. Data are collected during consultations, diagnostic tests, and routine monitoring, with risk factors being assessed during check-ups at a frequency determined by clinical guidelines and patient needs. The project protocol (no: 21_000431) was approved by the Independent Scientific Advisory Committee. Data from CPRD was linked to the Hospital Episode Statistics Admitted Patient Care (HES-APC), death registrations from the Office for National Statistics (ONS), and the Patient Level Index of Multiple Deprivation (IMD) 2019.

Patients were eligible to be included in the cohort if they were registered in the CPRD GOLD on the 1st of March 2017 (start date) with at least 1-year prior up-to-standard registration. Patients were eligible if they (1) were over 18 years of age on the start date, and (2) could be linked with HES-APC and ONS. From the eligible population, three separate cohorts were constructed by identifying individuals with diagnoses of T2DM, CVD (classified as a diagnosis of either coronary heart disease, stroke, peripheral arterial disease, or aortic disease), or CKD using both Read codes (CPRD) and ICD-10 codes (HES-APC). The three cohorts were not mutually exclusive as some individuals could have more than one of the three conditions.

Within each cohort, three time periods were constructed: 1st March 2018 – 28th February 2019; 1st March 2019 – 29th February 2020; and 1st March 2020 – 28th February 2021; hereafter referred to as 2018, 2019, and 2020, respectively. Each of the three cohorts included only those with the prevalent LTC for the period, i.e., those who had their LTC diagnosis on or before the start date (1st March) of the specific period. Individuals were included if they were alive and registered at the start of the period and excluded if any of the following occurred on or before the start date of the period: date of death, date of transferring out of the practice, or date of the practice's last collection of data. A graphical illustration of the cohort construction through inclusion or exclusion criteria is given in S1 Fig. The total follow-up period was from March 2018 to March 2021 (the end of the available linkage coverage).

### Risk factors and patient characteristics

RFs considered were systolic (SBP) and diastolic (DBP) blood pressure (BP), total cholesterol (TC), high-density lipoproteins (HDL), low-density lipoproteins (LDL), body mass index (BMI), smoking status, and HbA1c (for the T2DM cohort only); information was collected from CPRD records. Age, sex, ethnicity, and deprivation data were also collected for all the individuals in our study. Age was calculated at the beginning of each study period and grouped into two categories (<65 and ≥ 65 years). Sex was recorded as male or female. Ethnicity was primarily collected from HES; missing ethnicities were obtained from CPRD, if available. Ethnicity was categorised into four groups: White, South Asian, Black, and Mixed/Other.

Quintiles of IMD were used to quantify area-level socioeconomic deprivation. The IMD is a measure used by the UK government that considers various factors (income, employment, education, health, crime, barriers to housing and services, and the living environment) to assess the deprivation level in different areas [17]. Quintiles 1 and 5 represented the most and least deprived areas, respectively. Prevalent hypertension for each cohort was obtained from both CPRD and HES records.

## Statistical analysis

Demographic and clinical characteristics for each cohort were obtained using the most recent measure within a year before the start date of each period (1st March 2018, 1st March 2019, and 1st March 2020) and reported as either number (percentage [%]) for categorical variables or mean (standard deviation [SD]) or median (interquartile range [IQR]) for continuous variables.

Within each of the three cohorts, the proportions of individuals with at least one valid record of the selected RFs during each period were calculated and plotted. Among individuals whose RFs were assessed, the recorded values of each RF were pooled to calculate and plot the mean values and the proportions of individuals who had their RFs controlled at each period. The targets used for each RF were taken from NICE guidelines [18–20] (S1 Table). In case of multiple recordings, the last recorded value was selected. We explored these results further by stratifying the analysis by age, sex, ethnicity, and deprivation. As a sensitivity analysis, we have restricted the cohort to individuals with complete follow-up for each year and calculated the proportions of individuals with at least one valid record of the selected RFs.

As a secondary analysis, we used complete case univariable and multivariable logistic regression models to estimate the association of patient characteristics with the outcome variable of a patient having all the selected RFs assessed during a given period (yes vs no). These analyses were performed for the three cohorts and the three time periods separately to explore the temporal trends and differences across the LTCs. Model results are shown as odds ratios (OR) with 95% confidence intervals (95% CI). Age, sex, ethnicity, deprivation, and prevalent comorbidity (T2DM, CVD, CKD, and hypertension) were included as hypothesised predictors.

All analyses were conducted by using Python version 3.11.5, and STATA version 17.0.

## Results

### Characteristics of the three cohorts

Within the cohort of 769,551 patients identified in CPRD GOLD who met the study eligibility criteria (S2 Fig), 59,169 (7.7%) had a diagnosis of T2DM, 49,754 (6.5%) had a diagnosis of CVD, and 39,803 (5.2%) had a diagnosis of CKD at the start of 1st March 2018. Follow-up duration and missing values for baseline RFs are given in S2 Table.

In 2018, the T2DM cohort had a mean (SD) age of 64.0 (15.2) years, 51.4% were male, 80.6% were White, and 19.1% resided in the most socioeconomically deprived areas (Table 1). In comparison, the CVD cohort was older with a mean age of 71.9 (13.0) years. A higher proportion of individuals in the cohort were male (58.3%) and White (91.3%) while a lower proportion lived in the most deprived areas (16.6%) (Table 2). Those with a CKD diagnosis were the oldest on average among the three cohorts (mean age of 76.3 (11.4) years). The cohort also had the lowest proportion of males (45.5%), a high proportion of White individuals (88.2%), and the lowest proportion living in the most deprived areas (13.5%) (Table 3). The mean duration of diagnosis was highest in those with CVD (9.2 [SD, 7.9] years) in 2018 compared to those with T2DM and CKD (8.0 [SD, 6.5] years and 7.7 [SD, 3.9] years, respectively)

Each of the three cohorts reduced in size during the follow-up period, attributable to a higher rate of individuals being excluded from the cohorts compared to the incidence of new

**Table 1. Characteristics of people with type 2 diabetes.**

| Characteristics | Eligible cohort on 1st March 2018 (N = 59,169) | Eligible cohort on 1st March 2019 (N = 51,747) | Eligible cohort on 1st March 2020 (N = 38,256) |
|---|---|---|---|
| Age at start date (years), mean (SD) | 64.0 (15.2) | 64.4 (15.0) | 64.4 (14.9) |
| Age at start date (years), mean (SD) | | | |
| 18-44 | 6341 (10.7) | 5202 (10.1) | 3835 (10.0) |
| 45-64 | 22307 (37.7) | 19519 (37.7) | 14587 (38.1) |
| 65-74 | 14786 (25.0) | 12957 (25.0) | 9470 (24.8) |
| ≥75 | 15735 (26.6) | 14069 (27.2) | 10364 (27.1) |
| Age at T2DM diagnosis (years), mean (SD) | 55.7 (14.7) | 55.8 (14.6) | 55.7 (14.6) |
| Duration of T2DM (years), mean (SD) | 8.0 (6.5) | 8.2 (6.6) | 8.4 (6.7) |
| Sex, n (%) | | | |
| Male | 30419 (51.4) | 26606 (51.4) | 19550 (51.1) |
| Female | 28750 (48.6) | 25141 (48.6) | 18706 (48.9) |
| Race/Ethnicity, n (%) | | | |
| White | 47717 (80.6) | 42105 (81.4) | 30797 (80.5) |
| South Asian | 3459 (5.8) | 2636 (5.1) | 2310 (6.0) |
| Black | 1895 (3.2) | 1419 (2.7) | 1019 (2.7) |
| Mixed/Other | 2567 (4.3) | 2215 (4.3) | 1686 (4.4) |
| Missing | 3531 (6.0) | 3372 (6.5) | 2444 (6.4) |
| IMD (quintiles), n (%) | | | |
| 1 (least deprived) | 13645 (23.1) | 11828 (22.9) | 7900 (20.7) |
| 2 | 11696 (19.8) | 10287 (19.9) | 7209 (18.8) |
| 3 | 11903 (20.1) | 10557 (20.4) | 7889 (20.6) |
| 4 | 10579 (17.9) | 8955 (17.3) | 7169 (18.7) |
| 5 (most deprived) | 11330 (19.1) | 10108 (19.5) | 8083 (21.1) |
| Missing | 16 (0.0) | 12 (0.0) | 6 (0.0) |
| Smoking status, n (%) | | | |
| Smoker | 8721 (14.7) | 8010 (15.5) | 6053 (15.8) |
| Non-smoker | 19116 (32.3) | 16729 (32.3) | 12100 (31.6) |
| Ex-smoker | 14578 (24.6) | 13336 (25.8) | 9337 (24.4) |
| Missing | 16754 (28.3) | 13672 (26.4) | 10766 (28.1) |
| HbA1c at baseline (%), median [IQR] | 6.7 [6.1,7.6] | 6.6 [6.1,7.5] | 6.6 [6.0,7.5] |
| HbA1c at baseline (mmol/mol), median [IQR] | 49.7 [43.2,59.6] | 48.6 [43.2,58.5] | 48.6 [42.1,58.5] |
| Blood pressure (mm Hg), median [IQR] | | | |
| Diastolic | 77 [70,80] | 76 [70,80] | 77 [70,80] |
| Systolic | 132 [122,140] | 131 [122,140] | 131 [122,139] |
| BMI (kg/m²), mean (SD) | 29.7 [26.1,34.2] | 29.6 [26.0,34.1] | 29.6 [26.0,34.0] |
| BMI (kg/m²), mean (SD) | | | |
| <18.5 | 381 (1.0) | 317 (0.9) | 277 (1.1) |
| 18.5-24.9 | 6610 (17.2) | 5923 (17.6) | 4359 (17.8) |
| 25-29.9 | 12755 (33.2) | 11308 (33.6) | 8118 (33.1) |
| 30-34.9 | 10234 (26.6) | 8862 (26.3) | 6455 (26.4) |
| ≥35 | 8433 (22.0) | 7243 (21.5) | 5282 (21.6) |
| Total cholesterol (mmol/L), median [IQR] | 4.3 [3.6,5.1] | 4.2 [3.6,5.0] | 4.2 [3.6,5.0] |
| HDL (mmol/L), median [IQR] | 2.7 [2.1,3.5] | 2.7 [2.0,3.4] | 2.7 [2.1,3.5] |
| LDL (mmol/L), median [IQR] | 1.2 [1.0,1.5] | 1.2 [1.0,1.5] | 1.2 [1.0,1.5] |
| Comorbidities | | | |

*(Continued)*

**Table 1.** (Continued)

| Characteristics | Eligible cohort on 1st March 2018 (N = 59,169) | Eligible cohort on 1st March 2019 (N = 51,747) | Eligible cohort on 1st March 2020 (N = 38,256) |
|---|---|---|---|
| Cardiovascular Diseases (CVD) | 14839 (25.1) | 13357 (25.8) | 10076 (26.3) |
| Chronic Kidney Disease (CKD) | 11872 (20.1) | 10173 (19.7) | 6955 (18.2) |
| Hypertension | 36262 (61.3) | 31973 (61.8) | 23220 (60.7) |

T2DM: type 2 diabetes mellitus; IMD: Index of Multiple Deprivation; HbA1c: glycated haemoglobin BMI: body mass index; SD: standard deviation; IQR: interquartile range. HDL: high-density lipoprotein; LDL: low-density lipoprotein.

diagnoses within the CPRD records. The demographics of the three cohorts were relatively stable over the follow-up period with some minor exceptions. The cohorts experienced an overall increase in age and the proportions in the most deprived quintile. In the CKD cohort, the mean duration of the condition increased by one year to 8.7 years (SD, 4.5) and the proportion of males increased to 49.1% in 2020.

Comparing the three time periods, baseline RFs across all three cohorts remained relatively stable pre- and post-pandemic (Tables 1–3). In terms of prevalent conditions, among those with T2DM, 25.1% and 20.1% had CVD and CKD, respectively, in 2018. Among those with CVD, 29.8% and 28.8% had T2DM and CKD, respectively. In the CKD cohort, 29.8% and 36.0% had T2DM and CVD, respectively. The CKD cohort had the highest prevalence of hypertension (78.0%) compared with the T2DM (61.3%) and the CVD cohort (73.8%) in 2018.

## Risk factor assessment

Within the T2DM cohort, 66.4% and 65.2% had at least one recorded value of HbA1c in 2018 and 2019, which reduced to 43.4% in 2020 (Fig 1). The proportions of patients who had BP assessments were 74.0% in 2018 and 71.0% in 2019, which reduced to 40.7% in 2020. For TC assessments the proportion was 63.1% in 2018 and 60.3% in 2019, but it dropped significantly to 37.1% in 2020. BMI assessment decreased to 29.7% in 2020 from 57.3% and 55.0% in 2018 and 2019 respectively.

In those with CVD, 40.6% and 30.7% had their BP and TC measurements in 2020 (compared to 76.9% and 72.6% for BP, and 54.5% and 51.3% for TC, in 2018 and 2019 respectively). Furthermore, less than one in four (24.8%) patients had their BMI assessed in 2020 (which was 46.2% in 2018, and 44.1% in 2019)

In those with CKD, less than half of the patients (45.0%) had their BP assessed in 2020 while the proportions were 77.9% and 72.3% in 2018 and 2019, respectively. In 2020, TC and BMI assessment also reduced substantially to only 27.4% (51.8% in 2018 and 47.2% in 2019) and 23.5% (45.1% in 2018 and 42.0% in 2019).

The decrease in the assessment of LDL and HDL was similar across the three cohorts in 2020 (-21.9 and -22.5 percentage points (pp) difference from the previous year for T2DM; -19.5 and -19.7pp for CVD; -18.1 and -18.5pp for CVD). The proportion of individuals who had at least one record of their smoking status was also substantially lower in 2020 across all three cohorts (ranging from 32.0% to 38.0%) compared to previous years (ranging from 52.3 to 64.9%).

S3–S6 Figs report RFs assessment by age, sex, ethnicity, and deprivation, showing no differential patterns across sub-groups. The results of our sensitivity analysis (S7 Fig) with restricted cohorts (to those with complete follow up) were also consistent with our primary findings.

**Table 2. Characteristics of people with cardiovascular disease.**

| Characteristics | Eligible cohort on 1st March 2018 (N = 49,754) | Eligible cohort on 1st March 2019 (N = 43,048) | Eligible cohort on 1st March 2020 (N = 30,374) |
|---|---|---|---|
| Age at start date (years), mean (SD) | 71.9 (13.0) | 72.0 (12.8) | 72.0 (12.8) |
| Age at start date (years), mean (SD) | | | |
| 18-44 | 1324 (2.7) | 1034 (2.4) | 742 (2.4) |
| 45-64 | 12004 (24.1) | 10387 (24.1) | 7357 (24.2) |
| 65-74 | 13741 (27.6) | 11852 (27.5) | 8321 (27.4) |
| ≥75 | 22685 (45.6) | 19775 (45.9) | 13954 (45.9) |
| Age at CVD diagnosis (years), mean (SD) | 62.4 (14.0) | 62.4 (13.9) | 62.0 (13.9) |
| Duration of CVD (years), mean (SD) | 9.2 (7.9) | 9.4 (8.0) | 9.7 (8.2) |
| Sex, n (%) | | | |
| Male | 28994 (58.3) | 25229 (58.6) | 17637 (58.1) |
| Female | 20760 (41.7) | 17819 (41.4) | 12737 (41.9) |
| Race/Ethnicity, n (%) | | | |
| White | 45426 (91.3) | 39481 (91.7) | 27633 (91.0) |
| South Asian | 1441 (2.9) | 1076 (2.5) | 905 (3.0) |
| Black | 683 (1.4) | 512 (1.2) | 348 (1.1) |
| Mixed/Other | 1147 (2.3) | 985 (2.3) | 761 (2.5) |
| Missing | 1057 (2.1) | 994 (2.3) | 727 (2.4) |
| IMD (quintiles), n (%) | | | |
| 1 (least deprived) | 12544 (25.2) | 10381 (24.1) | 6637 (21.9) |
| 2 | 10333 (20.8) | 9081 (21.1) | 5996 (19.7) |
| 3 | 10121 (20.3) | 8919 (20.7) | 6395 (21.1) |
| 4 | 8480 (17.0) | 7215 (16.8) | 5414 (17.8) |
| 5 (most deprived) | 8264 (16.6) | 7443 (17.3) | 5926 (19.5) |
| Missing | 12 (0.0) | 9 (0.0) | 6 (0.0) |
| Smoking status, n (%) | | | |
| Smoker | 7819 (15.7) | 7050 (16.4) | 5255 (17.3) |
| Non-smoker | 12117 (24.4) | 10886 (25.3) | 7524 (24.8) |
| Ex-smoker | 13872 (27.9) | 12458 (28.9) | 8295 (27.3) |
| Missing | 15946 (32.0) | 12654 (29.4) | 9300 (30.6) |
| Blood pressure (mm Hg), median [IQR] | | | |
| Diastolic | 75 [69,80] | 75 [69,80] | 75.0 [70,80] |
| Systolic | 132 [122,140] | 131 [121,140] | 131.0 [121,140] |
| BMI (kg/m²), mean (SD) | 27.9 [24.6,31.9] | 27.8 [24.5,31.7] | 27.9 [24.5,31.9] |
| BMI (kg/m²), mean (SD) | | | |
| <18.5 | 548 (2.2) | 505 (2.2) | 364 (2.3) |
| 18.5-24.9 | 6378 (25.3) | 5935 (26.1) | 4184 (25.9) |
| 25-29.9 | 9265 (36.8) | 8293 (36.5) | 5789 (35.8) |
| 30-34.9 | 5656 (22.4) | 5060 (22.3) | 3676 (22.7) |
| ≥35 | 3358 (13.3) | 2913 (12.8) | 2161 (13.4) |
| Total cholesterol (mmol/L), median [IQR] | 4.2 [3.5,4.9] | 4.1 [3.4,4.8] | 4.0 [3.4,4.8] |
| HDL (mmol/L), median [IQR] | 2.5 [2.0,3.3] | 2.5 [1.9,3.2] | 2.5 [2.0,3.3] |
| LDL (mmol/L), median [IQR] | 1.3 [1.1,1.6] | 1.3 [1.0,1.6] | 1.3 [1.1,1.6] |
| Comorbidities | | | |
| Type 2 Diabetes Mellitus (T2DM) | 14839 (29.8) | 13357 (31.0) | 10076 (33.2) |
| Chronic Kidney Disease (CKD) | 14322 (28.8) | 12262 (28.5) | 8113 (26.7) |

*(Continued)*

**Table 2.** (Continued)

| Characteristics | Eligible cohort on 1st March 2018 (N = 49,754) | Eligible cohort on 1st March 2019 (N = 43,048) | Eligible cohort on 1st March 2020 (N = 30,374) |
|---|---|---|---|
| Hypertension | 36732 (73.8) | 32050 (74.5) | 22689 (74.7) |

CVD: cardiovascular diseases; IMD: Index of Multiple Deprivation; HbA1c: glycated haemoglobin BMI: body mass index; SD: standard deviation; IQR: interquartile range. HDL: high-density lipoprotein; LDL: low-density lipoprotein.

## Risk factor control

In the T2DM cohort, among those who had RF measurements, no substantial changes in the mean recorded values of HbA1c, BMI, or TC were observed during the pandemic compared to the previous year. SBP (+1.65 mmHg) and DBP (+1.02 mmHg) were higher in 2020 (Fig 2), resulting in a lower proportion of individuals with RF control (-4.6 percentage points (pp) and -5.2pp for SBP and DBP respectively) (Fig 3).

The mean recorded values of BMI remained unchanged for the CVD cohort, however, TC was slightly lower in 2020 (-0.03mmol/L) compared to 2019. SBP was higher by 1.30 mmHg and DBP by 0.67 mmHg, reducing the proportion of individuals with controlled BP, with decreases of 3.5pp for SBP and 2.5pp for DBP, respectively.

BMI was slightly lower (-0.31 kg/m$^2$) in the CKD cohort in 2020 compared to 2019, which improved the proportion who had it controlled (from 65.3% to 67.3%). Mean values of blood lipids was also lower (-0.13, -0.09, -0.03 mmol/L in TC, HDL, and LDL, respectively, in 2020 from 2019) resulting in a substantial increase in those who had their TC controlled (+5.1pp).

In all three cohorts, those who had their smoking status assessments, an increased proportion of patients reported smoking; the percentage of non-smokers or ex-smokers changed by -4.3, -5.5, and -5.1pp in the T2DM, CVD, and CKD cohorts, respectively. The mean values and patterns of RF control were largely consistent across age, sex, ethnicity, and deprivation (S8–S15 Figs).

## Factors associated with risk factor assessments

The associations of age, sex, ethnicity, deprivation, and prevalent comorbidity with the outcome of having all of the RFs measured during a particular year versus some or none of them measured are shown in Fig 4. The number of events and the univariable (unadjusted) results are reported in S3 Table.

Relative to the two years before the pandemic, in those with T2DM, the probability of having all RFs measured during the pandemic decreased in the elderly patients, particularly those aged ≥ 75 years (compared to those aged 18-44 years) (OR, 2.32, 95%CI: 1.96, 2.73 in T2DM in 2020; 2.91 (2.67, 3.18) in 2018), and male (compared to female) (1.30 (1.22, 1.37) in 2020; 1.37 (1.32, 1.42) in 2018) (Fig 4). Compared to the least deprived quintile, the association in the most deprived quintile was stronger in 2019 but weaker in 2020 (1.39 (1.27, 1.52) in 2020; 1.73 (1.63, 1.83) in 2019; 1.22 (1.16, 1.29) in 2018). In those with CVD, the association reduced for those elderly during the pandemic (OR: 1.64, 95%CI: 1.13, 2.37 in 2020; 2.16 (1.81, 2.57) in 2018 in those aged ≥ 75 years). In those with CKD, the trend was similar for elderly patients (0.94 (0.52, 1.72) in 2020; OR: 1.43 (1.05, 1.94) in 2018 in those aged ≥ 75 years). However, being male was associated with a higher probability (OR: 1.34, 95%CI: 1.21, 1.48 in 2020; 1.21 (1.15, 1.27) in 2018). Compared to the least deprived quintile the odds of RF measurement across all four quintiles (quintiles 2 to the most deprived quintile 5) increased over time for

**Table 3. Characteristics of people with chronic kidney disease.**

| Characteristics | Eligible cohort on 1st March 2018 (N = 39,803) | Eligible cohort on 1st March 2019 (N = 33,144) | Eligible cohort on 1st March 2020 (N = 20,680) |
|---|---|---|---|
| Age at start date (years), mean (SD) | 76.3 (11.4) | 76.6 (11.2) | 77.0 (11.1) |
| Age at start date (years), mean (SD) | | | |
| 18-44 | 384 (1.0) | 291 (0.9) | 168 (0.8) |
| 45-64 | 5460 (13.7) | 4419 (13.3) | 2678 (12.9) |
| 65-74 | 9731 (24.4) | 7983 (24.1) | 4724 (22.8) |
| ≥75 | 24228 (60.9) | 20451 (61.7) | 13110 (63.4) |
| Age at CKD diagnosis (years), mean (SD) | 68.4 (11.5) | 68.0 (11.4) | 67.9 (11.3) |
| Duration of CKD (years), mean (SD) | 7.7 (3.9) | 8.3 (4.1) | 8.7 (4.5) |
| Sex, n (%) | | | |
| Male | 18097 (45.5) | 15445 (46.6) | 10150 (49.1) |
| Female | 21706 (54.5) | 17699 (53.4) | 10530 (50.9) |
| Race/Ethnicity, n (%) | | | |
| White | 35101 (88.2) | 29262 (88.3) | 18091 (87.5) |
| South Asian | 825 (2.1) | 590 (1.8) | 455 (2.2) |
| Black | 970 (2.4) | 673 (2.0) | 407 (2.0) |
| Mixed/Other | 746 (1.9) | 624 (1.9) | 425 (2.1) |
| Missing | 2161 (5.4) | 1995 (6.0) | 1302 (6.3) |
| IMD (quintiles), n (%) | | | |
| 1 (least deprived) | 10908 (27.4) | 8952 (27.0) | 4981 (24.1) |
| 2 | 8945 (22.5) | 7539 (22.7) | 4465 (21.6) |
| 3 | 8118 (20.4) | 6886 (20.8) | 4416 (21.4) |
| 4 | 6458 (16.2) | 5306 (16.0) | 3595 (17.4) |
| 5 (most deprived) | 5369 (13.5) | 4459 (13.5) | 3222 (15.6) |
| Missing | 5 (0.0) | 2 (0.0) | 1 (0.0) |
| Smoking status, n (%) | | | |
| Smoker | 3674 (9.2) | 3309 (10.0) | 2322 (11.2) |
| Non-smoker | 11177 (28.1) | 9622 (29.0) | 6072 (29.4) |
| Ex-smoker | 10013 (25.2) | 8867 (26.8) | 5245 (25.4) |
| Missing | 14939 (37.5) | 11346 (34.2) | 7041 (34.0) |
| Blood pressure (mm Hg), median [IQR] | | | |
| Diastolic | 75 [68,80] | 75 [68,80] | 75 [69,80] |
| Systolic | 134 [124,140] | 133 [123,140] | 133 [123,140] |
| BMI (kg/m²), mean (SD) | 28.0 [24.6,32.0] | 27.8 [24.5,31.9] | 27.9 [24.7,31.9] |
| BMI (kg/m²), mean (SD) | | | |
| <18.5 | 353 (1.8) | 358 (2.1) | 224 (2.1) |
| 18.5-24.9 | 4995 (25.5) | 4364 (25.8) | 2699 (24.7) |
| 25-29.9 | 7204 (36.7) | 6260 (37.0) | 4060 (37.2) |
| 30-34.9 | 4363 (22.2) | 3686 (21.8) | 2370 (21.7) |
| ≥35 | 2695 (13.7) | 2261 (13.4) | 1555 (14.3) |
| Total cholesterol (mmol/L), median [IQR] | 4.4 [3.7,5.2] | 4.3 [3.6,5.1] | 4.2 [3.5,5.0] |
| HDL (mmol/L), median [IQR] | 2.8 [2.1,3.5] | 2.7 [2.1,3.5] | 2.6 [2.0,3.4] |
| LDL (mmol/L), median [IQR] | 1.3 [1.1,1.6] | 1.3 [1.1,1.6] | 1.3 [1.1,1.6] |
| Comorbidities | | | |
| Type 2 Diabetes Mellitus (T2DM) | 11872 (29.8) | 10173 (30.7) | 6955 (33.6) |
| Cardiovascular Diseases (CVD) | 14322 (36.0) | 12262 (37.0) | 8113 (39.2) |

*(Continued)*

**Table 3.** (Continued)

| Characteristics | Eligible cohort on 1st March 2018 (N = 39,803) | Eligible cohort on 1st March 2019 (N = 33,144) | Eligible cohort on 1st March 2020 (N = 20,680) |
|---|---|---|---|
| Hypertension | 31032 (78.0) | 25996 (78.4) | 16384 (79.2) |

CKD: Chronic Kidney Disease; IMD: Index of Multiple Deprivation; HbA1c: glycated haemoglobin BMI: body mass index; SD: standard deviation; IQR: interquartile range. HDL: high-density lipoprotein; LDL: low-density lipoprotein.

those with CVD or CKD. For different ethnic categories, no substantial change was observed among the three cohorts.

## Discussion

### Summary of our findings

Our analysis shows that the assessments of several key RFs among people with cardiometabolic diseases (T2DM, CVD, or CKD) decreased significantly during the pandemic. This reduction was observed consistently across all three conditions, all RFs examined, and in stratified analyses by age, sex, ethnicity, and deprivation. Additionally, we found an increase in BP levels among individuals with T2DM, leading to a higher proportion of patients with uncontrolled BP during the pandemic. During the pandemic, the likelihood of older patients having all RFs assessed compared to younger patients decreased in all three cohorts. Among those with T2DM, the likelihood of males having all RFs assessed compared to females decreased, whereas in the CKD cohort, males showed a stronger likelihood in 2020.

### Findings in context

To our knowledge, our study is the first to investigate the effect of the pandemic on health checks for those with major long-term conditions. Although previous studies have explored changes in RFs, these studies were either based on the overall population or only one LTC was considered. Furthermore, to the best of our knowledge, no other study has conducted a comprehensive analysis on those with T2DM, CVD and CKD. This research not only highlights the specific impact on a vulnerable segment of the population but also provides insights that could inform more targeted health policy interventions even in the face of global health crises.

In a previous study among people with diabetes using the UK National Diabetes Audit (NDA), a 45% relative reduction in the proportion of people receiving all eight annual care processes (including HbA1c, BP, cholesterol, BMI, and smoking status) was observed in 2020–21 compared to 2019–20, with an increased rate of non-COVID-19-related mortality in those who did not receive all eight care processes [21]. Another study among people with T2DM showed that during April 2020 rates of performing health checks (including HbA1c, cholesterol, blood pressure, and BMI) were reduced by 76%–88% when compared with 10-year historical trends, with older people from deprived areas experiencing the greatest reductions; the rates improved by the end of 2020 but remained at 28%–47% [22]. A study conducted in the United States also showed significant declines in both outpatient visits and HbA1c testing among adults with T2DM in the early stages of the pandemic. Nevertheless, HbA1c levels showed relative stability in 2020 compared to 2019, which was also confirmed by our study [23]. Another study using testing data from ten UK Clinical Biochemistry Departments showed that monthly HbA1c requests decreased by more than 80% in April 2020 compared to the average monthly request numbers in 2019 [24].

**a) T2DM Cohort**

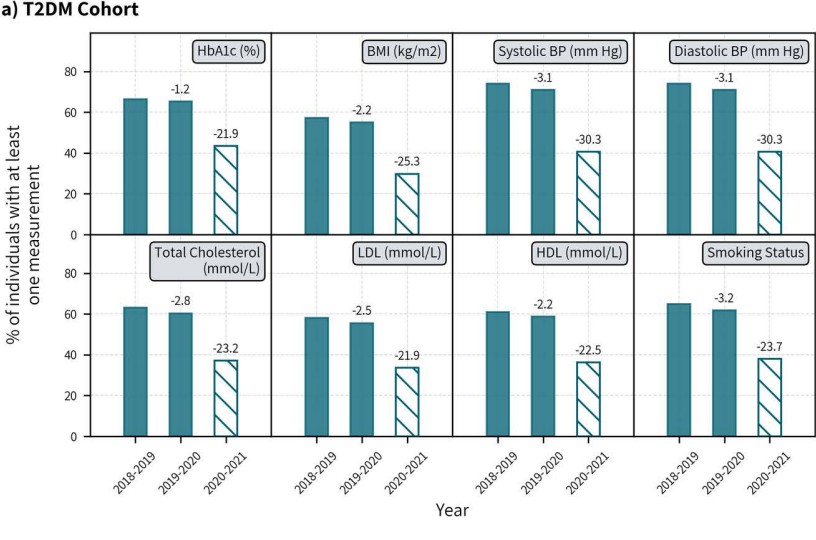

**b) CVD Cohort**

**c) CKD Cohort**

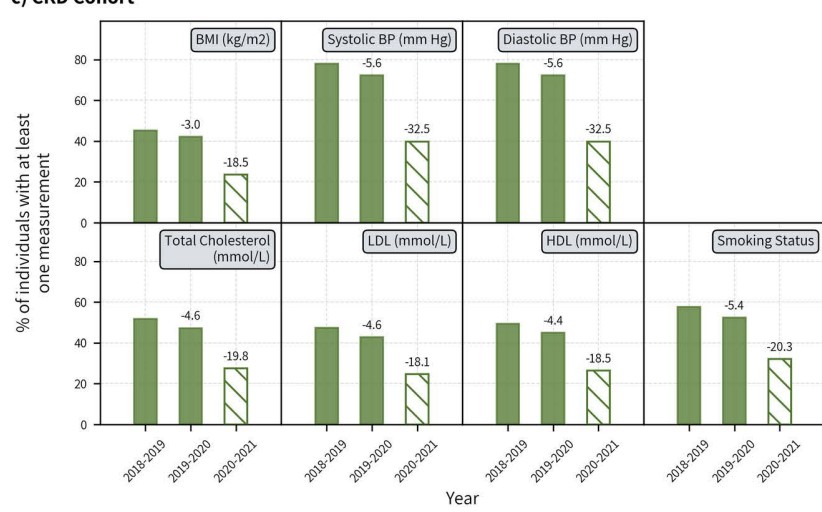

**Fig 1. Proportion of individuals in the T2DM, CVD, and CKD cohort who had at least one measurement of the selected risk factors recorded in CPRD during the pre-pandemic period of 2018-2019 and 2019-2020, and the**

**pandemic period of 2020-2021.** *Numbers above bars represent percentage point (pp) change from the previous year. Solid bars represent pre-pandemic years and hashed bar represents the pandemic year.

The changes in BP assessments and increase in SBP and DBP during the pandemic has also been reported by several previous studies [25–32]. A study from the United States found that BP measurements substantially dropped early in the pandemic for hypertensive patients and SBP and DBP increased by 1.79 mmHg (95% CI: 1.57-2.01) and 1.30 mmHg (1.18-1.42), respectively, compared to the pre-pandemic period [25]. The Cardiovascular Disease Prevention Audit (CVDPREVENT) found a 21.4% reduction in BP measurements among hypertensive patients [28]. Another study using primary care data in England reported a 42% reduction in BP measurement between April 2019 - April 2021 [29]. Another study from the United States showed that, during primary care visits, measurements of BP decreased by 50% and cholesterol levels by 37% in the second quarter of 2020 compared with 2018-2019 levels [30]. Furthermore, a Canadian study also reported a significant reduction (76% to 36%) in BP documentation during hypertension visits from the pre-pandemic to the pandemic period [31]. Our study showed similar changes in BP levels persisted across those with major LTCs which is concerning as even a 2-5 mm Hg reduction in BP has been associated with significantly lower cardiovascular events and death [32].

Compared to pre-pandemic trends, previous studies also found reduced rates of BMI and lipids measurements, improvements in lipid profile, and increased prevalence of smoking during 2020 [22,33–39]. We observed minimal change in BMI in our study in 2020 compared to the preceding two years but previous studies reported both weight loss and gain during the lockdowns [40,41]. The difference was largely because we reported the mean BMI for the overall cohort, whereas the change reported in previous studies varied between subcategories of baseline BMI [40,42]. Another study using primary care records in England found twelve key RF measurements experienced a sharp decline in March 2020, but most rebounded to the lower bounds of expected levels by 2022, with the exception of BP and HbA1c, which remained below expected levels [43]. Our study further highlights the disproportionate impact of these disruptions across cohorts, showing that individuals with LTCs experienced significant gaps in RF assessments during the same period.

The reduced likelihood of RF assessments in older patients in 2020 are particularly concerning given their higher baseline vulnerability to complications from poorly managed chronic conditions. Studies have highlighted that delays in care for older adults during the pandemic were associated with worsening disease control and increased mortality risks, particularly for those with multiple chronic conditions [44]. The observed sex-specific differences in the trend of the likelihood of RF assessments between the T2DM and CKD cohorts may stem from variations in healthcare utilisation within these conditions [45].

## Explanation of our findings

Changes in RF management during the pandemic raise several important points for discussion. Firstly, the sub-group within each cardiometabolic LTC that had their RFs recorded in 2018 and 2019 may have changed in composition in 2020. Before the pandemic, patients suspected of being at high risk for developing further complications might have been more likely to have routine check-ups scheduled by General Practitioners (GPs) for continuous assessments. GPs who were over-stretched managing the influx of COVID-19 cases may have shifted their attention to individuals with non-routine (acute) care and urgent needs [46]. Furthermore, routine healthcare visits were deferred as there were concerns about shortages

**a) T2DM Cohort**

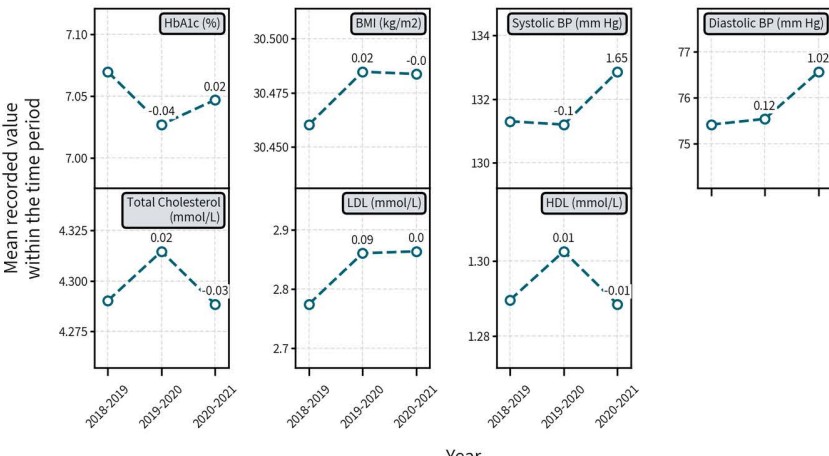

**b) CVD Cohort**

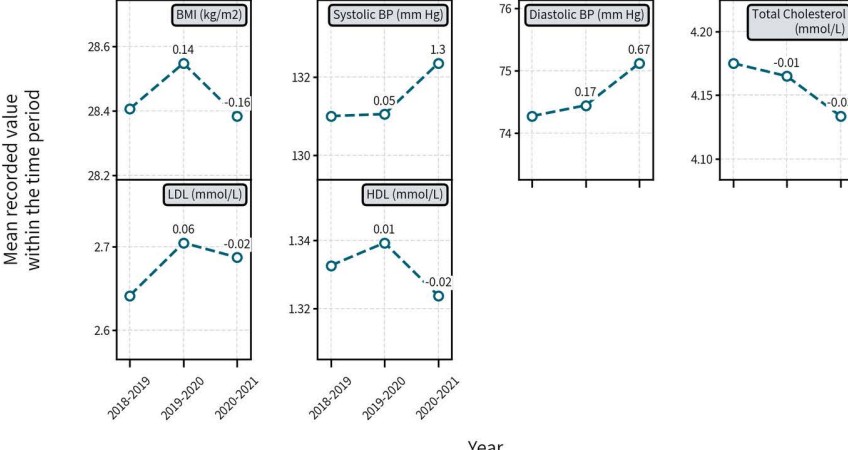

**c) CKD Cohort**

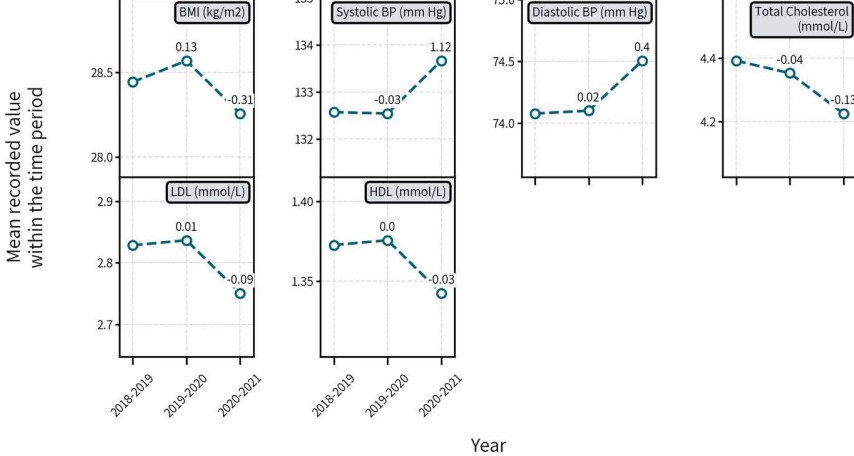

**Fig 2. Mean recorded values of selected risk factors among those with at least one measurement in CPRD in the T2DM, CVD, and CKD cohort during the pre-pandemic period of 2018-2019 and 2019-2020, and the pandemic period of 2020-2021.** *Numbers above circles represent changes in absolute values from the previous year.

### a) T2DM Cohort

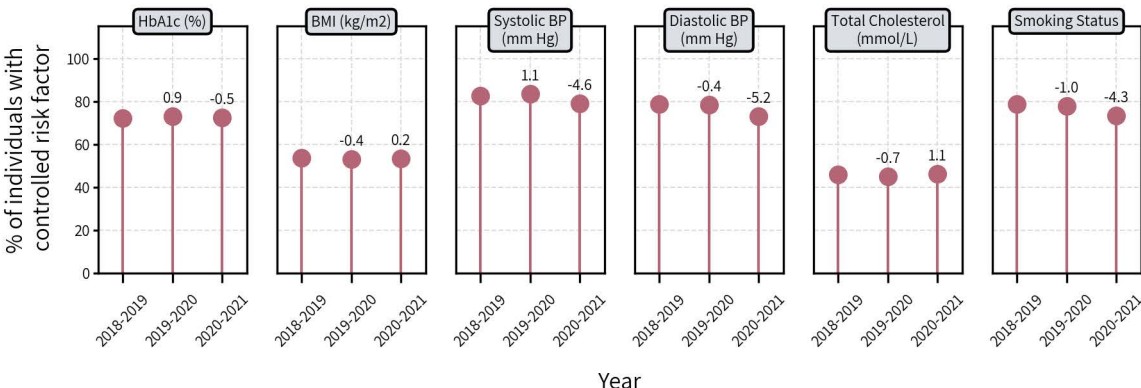

### b) CVD Cohort

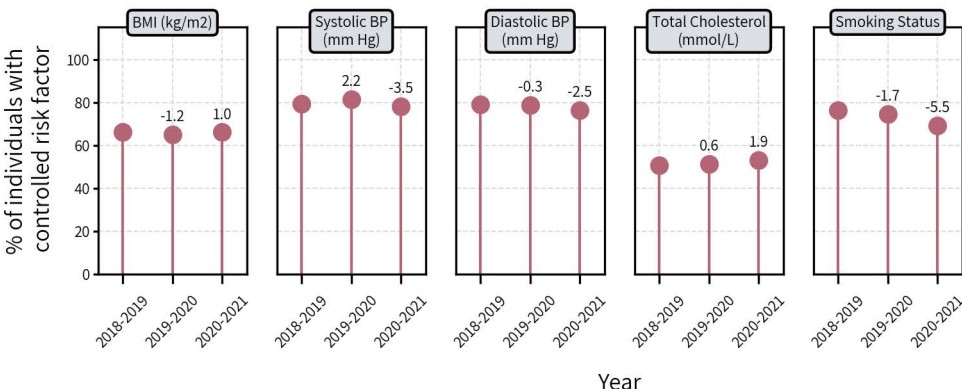

### c) CKD Cohort

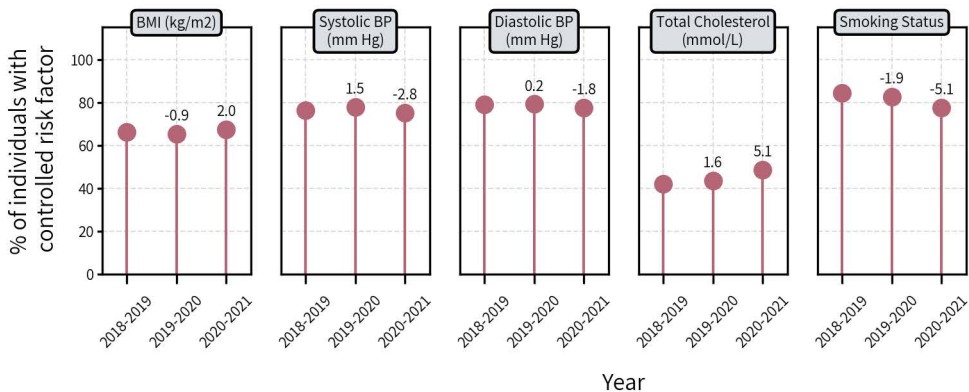

**Fig 3. Proportion of individuals who had their risk factors controlled among those with at least one measurement in CPRD in the T2DM, CVD, and CKD cohort during the pre-pandemic period of 2018-2019 and 2019-2020, and the pandemic period of 2020-2021.** *Numbers above circles represent percentage point (pp) change from the previous year.

of resources, staffing, and personal protective equipment. Consequently, during the pandemic high-risk individuals were advised to stay at home. All of these may have resulted in fewer recording of RFs and different sub-groups appearing for RF recordings.

Social distancing and lifestyle changes during the pandemic, may have influenced patient behaviours including diet and physical activity [39,47,48]. These changes, including an increase in pandemic-related depression, stress, anxiety, and boredom [49] could have impacted BP and smoking, leading to increases in both across all three LTCs.

There may have also been a change in how the RFs were recorded. As the lockdown began, healthcare providers increasingly adopted telemedicine (remote consultation services). Instead of being taken by a GP at a face-to-face appointment, some recordings may have been taken by patients at home and communicated over the phone. Additionally, the relaxation of lockdown measures during the second half of 2020, switching to regional tier systems, and the rollout of COVID-19 vaccines to the public in December 2020 could have further affected healthcare management patterns for patients with LTCs.

## Implications for policy

RF monitoring in primary care for those with LTCs is essential for efficient, proactive, and personalised healthcare. For instance, with the absence of regular testing, individuals with T2DM may experience delayed detection of poor glycaemic control which can compromise the clinical decision-making, and lead to worsening of already existing therapeutic inertia (i.e., delay or reluctance in intensifying treatment despite the persistence of suboptimal glycaemic control), and fail to prevent mortality, and long-term micro- and macrovascular outcomes of T2DM [9,10,50]. Tight risk factor control including blood pressure and lipid control is also essential to mitigate life-threatening CVD and kidney events, including heart failure, stroke, and renal failure; the risks of which are particularly elevated among those with T2DM, CVD, or CKD [51,52]. Additionally, considering the significant association between smoking cessation and reduced risks of cardiovascular disease, CKD progression, and mortality [53,54], stronger interventions need to be implemented in future to avoid long-term consequences. To address risk factor management disruptions, healthcare policies must prioritise the integration of resilient systems capable of maintaining regular monitoring and management of high-risk groups, even during crises. Specifically, increasing the use of telemedicine, home monitoring tools, and targeted outreach programs would ensure continuity of care and prevent worsening outcomes in vulnerable populations. Additionally, integrating disaster medicine specialty into healthcare planning can improve preparedness and response during emergencies. Our study provides evidence to inform policies aimed at strengthening primary care infrastructure, and implementing proactive measures to mitigate disparities in chronic disease outcomes during future disruptions.

## Strengths and limitations

Our study utilising a large cohort from CPRD offers several strengths. CPRD provides a vast and representative dataset derived from real-world electronic healthcare records, capturing a diverse patient population across the United Kingdom. Its longitudinal nature allows for the comparison of pre-pandemic clinical records with those from the pandemic period, offering insights into the trends over time. Unlike other studies, we examined all key cardiometabolic conditions most affected by the pandemic (T2DM, CV and CKD) and assessed not only measurement of risk factors but also their outcomes. Additionally, we stratified our analysis by age, sex, ethnicity and deprivation. The results therefore have major implications for policymakers.

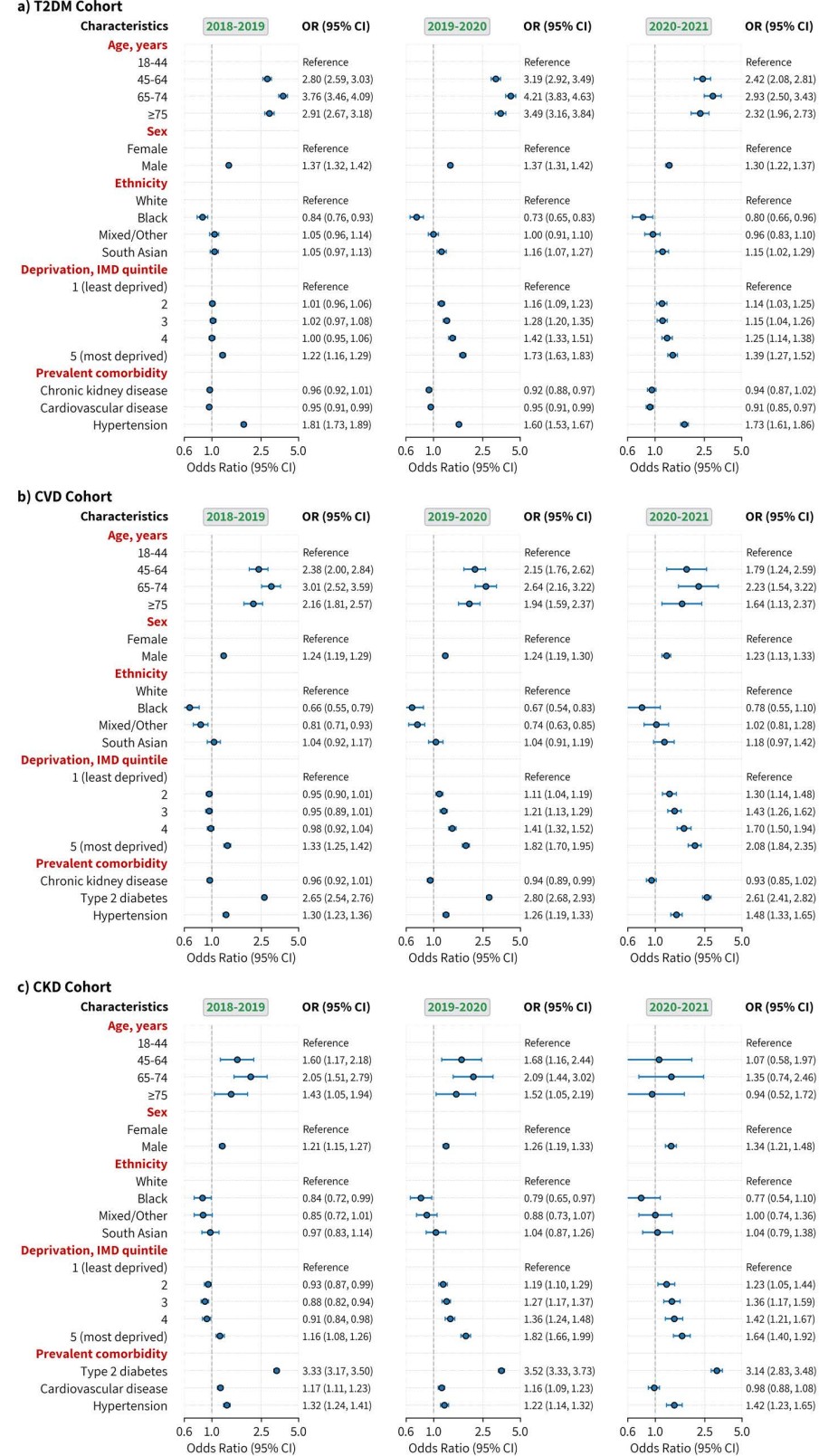

**Fig 4. Adjusted odds ratios for having all of the risk factors assessed for individuals in the T2DM, CVD, and CKD cohort during the pre-pandemic period of 2018-2019 and 2019-2020, and the pandemic period of**

**2020-2021.** Complete case multivariable logistic regression models, mutually adjusted for all variables shown in the plot, were used to estimate the association between patient characteristics (age, sex, ethnicity, deprivation, and prevalent condition (T2DM, CVD, CKD, and hypertension)) with the binary outcome variable of a patient having all the selected risk factors (systolic (SBP) and diastolic (DBP) blood pressure (BP), total cholesterol (TC), high-density lipoproteins (HDL), low-density lipoproteins (LDL), body mass index (BMI), smoking status, and HbA1c (for the T2DM cohort only)) measured during a given period. For prevalent conditions, the reference is the absence of the condition. OR=odds ratio, IMD=indices of multiple deprivation, T2DM=type 2 diabetes, CVD=cardiovascular disease, CKD=chronic kidney disease, CI=Confidence Intervals.

A key limitation of our study was the potential variability in RF measurement and recording in routine primary care because of systematic biases and changes in medical decisions in practice over time. These changes, along with differences between actual and recorded dates of assessments, must be accounted for when interpreting trends in RFs. Also, if fewer people were seeing their GP during the pandemic, fewer people were getting diagnosed with new diseases. The same factors affecting RF control could also affect disease diagnosis and the number of patients in the cohort. Additionally, we only have data on measurements in those who had a record - those at the highest risks may have worse control but we do not have that data. Although we accounted for age, sex, ethnicity, and deprivation in our regression models, we acknowledge that biases cannot be entirely eliminated, as residual confounders may remain. Unmeasured factors, such as behavioural influences, healthcare access disparities, or policy changes, could affect the observed associations and trends.

Additionally, the results of our study using CPRD data in the UK may not be fully generalisable outside the UK as the demographics, clinical practices, access to healthcare, treatment protocols, medication availability, and response to the pandemic can vary widely between countries. Finally, our analyses were descriptive with a limited follow-up time and exploring the causal impact of COVID-19 on health checks is out of the scope of our study. Future studies with more follow-up time are also required to explore the consequences of such disruptions on long-term complications.

## Conclusion

In conclusion, our study highlights a substantial reduction in the assessments of several key RFs and poor control in those assessed during the pandemic among people with major long-term conditions. These findings underscore critical gaps in preventative care and chronic disease management during public health emergencies. To address these issues, healthcare policies must prioritise the integration of resilient systems capable of maintaining regular monitoring and management of high-risk groups during acute crises.

## Supporting information

**S1 Table. Target measurements for defining controlled risk factors.**
(PDF)

**S2 Table. Follow-up duration and missing values for baseline risk factors.**
(PDF)

**S3 Table. Unadjusted odds ratios for having at least one record of all risk factors for individuals in the T2DM, CVD, and CKD cohort during the pre-pandemic period of 2018-2019 and 2019-2020, and the pandemic period of 2020-2021.**
(PDF)

**S1 Fig. Cohort definition sample diagram.**
(PDF)

**S2 Fig. Study population flow diagram.**
(PDF)

**S3 Fig. Proportion of individuals, by age groups, in the T2DM, CVD and CKD cohort who had at least one measurement of the selected risk factors recorded in CPRD during the pre-pandemic period of 2018-2019 and 2019-2020, and the pandemic period of 2020-2021.**
(PDF)

**S4 Fig. Proportion of individuals, by sex, in the T2DM, CVD and CKD cohort who had at least one measurement of the selected risk factors recorded in CPRD during the pre-pandemic period of 2018-2019 and 2019-2020, and the pandemic period of 2020-2021.**
(PDF)

**S5 Fig. Proportion of individuals, by ethnicity, in the T2DM, CVD and CKD cohort who had at least one measurement of the selected risk factors recorded in CPRD during the pre-pandemic period of 2018-2019 and 2019-2020, and the pandemic period of 2020-2021.**
(PDF)

**S6 Fig. Proportion of individuals, by deprivation, in the T2DM, CVD and CKD cohort who had at least one measurement of the selected risk factors recorded in CPRD during the pre-pandemic period of 2018-2019 and 2019-2020, and the pandemic period of 2020-2021.**
(PDF)

**S7 Fig. Proportion of individuals in the restricted (to those with complete follow-up) T2DM, CVD and CKD cohort who had at least one measurement of the selected risk factors recorded in CPRD during the pre-pandemic period of 2018-2019 and 2019-2020, and the pandemic period of 2020-2021.**
(PDF)

**S8 Fig. Mean of recorded values of selected risk factors among those with at least one measurement in CPRD in the T2DM, CVD, and CKD cohort during the pre-pandemic period of 2018-2019 and 2019-2020, and the pandemic period of 2020-2021, by age groups.**
(PDF)

**S9 Fig. Mean recorded values of selected risk factors among those with at least one measurement in CPRD, in the T2DM, CVD, and CKD cohort during the pre-pandemic period of 2018-2019 and 2019-2020, and the pandemic period of 2020-2021, by sex.**
(PDF)

**S10 Fig. Mean recorded values of selected risk factors among those with at least one measurement in CPRD in the T2DM, CVD, and CKD cohort during the pre-pandemic period of 2018-2019 and 2019-2020, and the pandemic period of 2020-2021, by ethnicity.**
(PDF)

**S11 Fig. Mean recorded values of selected risk factors among those with at least one measurement in CPRD in the T2DM, CVD, and CKD cohort during the pre-pandemic period of 2018-2019 and 2019-2020, and the pandemic period of 2020-2021, by deprivation.**
(PDF)

**S12 Fig. Proportion of individuals who had their risk factors controlled among those with at least one measurement in CPRD in the T2DM, CVD, and CKD cohort during the pre-pandemic period of 2018-2019 and 2019-2020, and the pandemic period of 2020-2021, by age groups.**

(PDF)

**S13 Fig. Proportion of individuals who had their risk factors controlled among those with at least one measurement in CPRD in theT2DM, CVD, and CKD cohort during the pre-pandemic period of 2018-2019 and 2019-2020, and the pandemic period of 2020-2021, by sex.**
(PDF)

**S14 Fig. Proportion of individuals who had their risk factors controlled among those with at least one measurement in CPRD in the T2DM, CVD, and CKD cohort during the pre-pandemic period of 2018-2019 and 2019-2020, and the pandemic period of 2020-2021, by ethnicity.**
(PDF)

**S15 Fig. Proportion of individuals who had their risk factors controlled among those with at least one measurement in CPRD in the T2DM, CVD, and CKD cohort during the pre-pandemic period of 2018-2019 and 2019-2020, and the pandemic period of 2020-2021, by deprivation.**
(PDF)

**S1 File.** The RECORD statement: Checklist of items, extended from the STROBE statement, that should be reported in observational studies using routinely collected health data.
(PDF)

**S1 Checklist.** Statements (RECORD Checklist).
(PDF)

## Acknowledgments

KK, ShS, FZ, CR, YC, and CLG are supported by the National Institute for Health Research (NIHR) Applied Research Collaboration East Midlands (ARC EM), NIHR Global Research Centre for Multiple Long-Term Conditions and the NIHR Leicester Biomedical Research Centre (BRC). This research used the ALICE High Performance Computing Facility at the University of Leicester.

## Author contributions

**Conceptualization:** Clare L Gillies, Kamlesh Khunti, Francesco Zaccardi.

**Data curation:** Sharmin Shabnam, Clare L Gillies.

**Formal analysis:** Sharmin Shabnam.

**Funding acquisition:** Clare L Gillies, Kamlesh Khunti.

**Methodology:** Clare L Gillies, Francesco Zaccardi, Sharmin Shabnam, Kamlesh Khunti.

**Supervision:** Clare L Gillies, Francesco Zaccardi, Nazrul Islam, Kamlesh Khunti.

**Visualization:** Sharmin Shabnam.

**Writing – original draft:** Sharmin Shabnam.

**Writing – review & editing:** Francesco Zaccardi, Tom Yates, Nazrul Islam, Cameron Razieh, Yogini Chudasama, Amitava Banerjee, Samuel Seidu, Kamlesh Khunti, Clare L Gillies.

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
