## [Decision Letter · Decision Letter 0]

2 Dec 2024

PONE-D-24-38823Risk factor measurement in the COVID-19 pandemic in individuals with cardio-renal-metabolic diseases: A retrospective cohort study in the United KingdomPLOS ONE

Dear Dr. Khunti,

Thank you for submitting your manuscript to PLOS ONE. After careful consideration, we feel that it has merit but does not fully meet PLOS ONE’s publication criteria as it currently stands. Therefore, we invite you to submit a revised version of the manuscript that addresses the points raised during the review process.

This retrospective study utilizes useful scientific data and have compared patients' clinical conditions pre- and peripandemic. 

Few things needs to be corrected:

The main objective and its conclusion of the study is not elusive. How the findings from the study could be recommended for policy making; Justify?

Data Analysis still needs to be refined:

Specific factors like practitioner and participant reasons for educed risk factor assessments can be incorporated.

Qualitative data like direct questions/interview can be helpful.

Patient Recruitment:

Data set has been taken from a specific population or has been included based on specific criteria needs to be elaborated.

Statistical Analysis:

Author should include multivariate regression and post hoc analysis like in case of gender and ethnic disparities.

Discussion:

Rephrase the first paragraph of discussion.

Discuss about the potential biases in the study. How have you neutralized the confounding factors?

How have you handled the missing data or loss to follow up?

Include more studies from recent literature and discuss the findings with your study.

We look forward to receiving your revised manuscript.

Kind regards,

Apeksha Niraula, M.D., Biochemistry

Academic Editor

PLOS ONE

Journal Requirements:

4. Thank you for stating the following financial disclosure: This research was part of the Data and Connectivity National Core Study, led by Health Data Research UK in partnership with the Office for National Statistics and funded by UK Research and Innovation (grant ref MC_PC_20058). This work was also supported by The Alan Turing Institute via ‘Towards Turing 2.0’ EPSRC Grant Funding.

5. Thank you for stating the following in your Competing Interests section:  KK has acted as a consultant, speaker or received grants for investigator-initiated studies for Astra Zeneca, Bayer, Novartis, Novo Nordisk, Sanofi-Aventis, Lilly and Merck Sharp & Dohme, Boehringer Ingelheim, Oramed Pharmaceuticals, Pfizer, Roche, Daiichi-Sankyo and Applied Therapeutics. All authors declare no conflict of interest. 

6. In the online submission form, you indicated that this study was conducted using CPRD GOLD and linked data subject to protocol approval. Protocol for this study is available on CPRD website: https://www.cprd.com/approved-studies/investigating-disruptions-primary-care-result-covid-19-pandemic-and-introduction. The data controller for CPRD (Department of Health and Social Care) does not allow sharing raw data with third parties. Researchers may apply for data access at: https://www.cprd.com/research-applications. Statistical code is available upon request from the first author. Phenotypes used to define the cohort, medical conditions, and ethnicity are available at the GitHub link:

https://github.com/shabnam-shbd/risk_factor_control_during_covid19.

Additional Editor Comments:

This retrospective study utilizes useful scientific data and have compared patients' clinical conditions pre- and peripandemic.

Few things needs to be corrected:

The main objective and its conclusion of the study is not elusive. How the findings from the study could be recommended for policy making; Justify?

Data Analysis still needs to be refined:

Specific factors like practitioner and participant reasons for educed risk factor assessments can be incorporated.

Qualitative data like direct questions/interview can be helpful.

Patient Recruitment:

Data set has been taken from a specific population or has been included based on specific criteria needs to be elaborated.

Statistical Analysis:

Author should include multivariate regression and post hoc analysis like in case of gender and ethnic disparities.

Discussion:

Rephrase the first paragraph of discussion.

Discuss about the potential biases in the study. How have you neutralized the confounding factors?

How have you handled the missing data or loss to follow up?

Include more studies from recent literature and discuss the findings with your study.

Reviewers' comments:

Reviewer's Responses to Questions

**Comments to the Author**

1. Is the manuscript technically sound, and do the data support the conclusions?

Reviewer #1: Yes

Reviewer #2: Yes

2. Has the statistical analysis been performed appropriately and rigorously? 

Reviewer #1: Yes

Reviewer #2: Yes

3. Have the authors made all data underlying the findings in their manuscript fully available?

Reviewer #1: Yes

Reviewer #2: Yes

4. Is the manuscript presented in an intelligible fashion and written in standard English?

Reviewer #1: No

Reviewer #2: Yes

5. Review Comments to the Author

Reviewer #1: The title requires rewriting.

In the Introduction section, state the risk factors of COVID 19 Disease.

The sampling method should be specified.

The discussion is very complicated and makes it heavier for the reader also needs to become stronger.

The conclusion is very weak and requires re-examination and rewriting.

Reviewer #2: This retrospective study offers valuable insights by analyzing a wealth of secondary data to compare patients' clinical conditions pre- and peripandemic. While the findings are well-presented, the study could benefit from a clearer application of results to inform future policymaking. Specifically:

Practical Implications:

Emphasize how findings translate into actionable recommendations for policy and practice, especially in preparation for future scenarios involving resource constraints.

Additional Data Considerations:

Incorporating specific factors or variables, such as practitioners’ and participants’ reasons for reduced risk factor assessments, could provide deeper insights.

Exploring concurrent qualitative data (e.g., interviews) would add context to the findings.

Database and Population Scope:

Clarifying whether the dataset includes all or only a specific part of the population and the criteria for selecting specific practices/hospitals would greatly enhance the visualisation of the method and findings

Explain how the sample represents the broader population and assessment frequency.

Analysis and Findings:

Including other such as multivariate and post-hoc analyses could provide clarity on observed differences (e.g., gender and ethnic disparities) and contextualize these within recent literature.

The study would benefit from using references less than 10 years old, ensuring relevance (e.g., updated diabetes prevalence in the UK).

Cohort Design and Limitations:

Discuss limitations, such as potential biases due to the cohort design compared to other such as cross-sectional or qualitative methods.

Address drop-out rates or missing data and their implications for findings.

Impactful Results Discussion:

Highlight significant and actionable results, ensuring these are fully elaborated to contextualize their importance and relevance to current issues.

By addressing these areas, the study's topic to relevance and contribution for both academic and policy discussions can be significantly enhanced.

6. PLOS authors have the option to publish the peer review history of their article (what does this mean? ). If published, this will include your full peer review and any attached files.

**Do you want your identity to be public for this peer review?** For information about this choice, including consent withdrawal, please see our Privacy Policy .

Reviewer #1: No

Reviewer #2: No

---

## [Author Response · Author response to Decision Letter 1]

30 Jan 2025

Additional Editor Comments:

This retrospective study utilizes useful scientific data and have compared patients' clinical conditions pre- and peripandemic.

Few things needs to be corrected:

The main objective and its conclusion of the study is not elusive. How the findings from the study could be recommended for policy making; Justify?

Authors’ responses:

Thank you for the valuable feedback.

As suggested, we have revised our study objective in the Introduction section to make it clearer and more concrete as below (this can also be viewed in the revised manuscript with tracked change):

This study aimed to analyse the impact of the COVID-19 pandemic on the assessment of key RFs in individuals with cardio-renal-metabolic LTCs, specifically type 2 diabetes (T2DM), cardiovascular disease (CVD), and chronic kidney disease (CKD). These conditions were chosen as they are highly prevalent (globally about 530 million people have diabetes [11]; in the United Kingdom – approximately 5.6 million people have diabetes, 90% of which are type 2 [12]; 6.4 million have CVD [13]; and 3.5 million have CKD stages 3-5 [14]) and they are some of the leading causes of death and disability worldwide. Using routinely collected healthcare data, we sought to identify changes in RF monitoring during the pandemic and evaluate differences by demographic and clinical characteristics to inform strategies for improving chronic disease management in future public health emergencies.

References

11. Ong KL, Stafford LK, McLaughlin SA, Boyko EJ, Vollset SE, Smith AE, et al. Global, regional, and national burden of diabetes from 1990 to 2021, with projections of prevalence to 2050: a systematic analysis for the Global Burden of Disease Study 2021. The Lancet. 2023;402:203–34.

12. How many people in the UK have diabetes? [Internet]. Diabetes UK. [cited 2025 Jan 8]. Available from: https://www.diabetes.org.uk/about-us/about-the-charity/our-strategy/statistics

13. Team BHI. BHF England CVD Factsheet. 2024;

14. Facts about kidneys | Kidney Care UK [Internet]. Available from: https://kidneycareuk.org/kidney-disease-information/about-kidney-health/facts-about-kidneys/

We have also revised the Conclusion to make it clearer and stronger (this can also be viewed in the revised manuscript with tracked change):

In conclusion, our study highlights a substantial reduction in the assessments of several key RFs and poor control in those assessed during the pandemic among people with major long-term conditions. These findings underscore critical gaps in preventative care and chronic disease management during public health emergencies. To address these issues, healthcare policies must prioritise the integration of resilient systems capable of maintaining regular monitoring and management of high-risk groups during acute crises.

As suggested, we have also added the below paragraph detailing the recommendations for policy making in the Discussion section under the sub-headline Implications for policy (this can also be viewed in the revised manuscript with tracked change):

To address risk factor management disruptions, healthcare policies must prioritise the integration of resilient systems capable of maintaining regular monitoring and management of high-risk groups, even during crises. Specifically, increasing the use of telemedicine, home monitoring tools, and targeted outreach programs would ensure continuity of care and prevent worsening outcomes in vulnerable populations. Additionally, integrating disaster medicine specialty into healthcare planning can improve preparedness and response during emergencies. Our study provides evidence to inform policies aimed at strengthening primary care infrastructure, and implementing proactive measures to mitigate disparities in chronic disease outcomes during future disruptions.

Data Analysis still needs to be refined:

Specific factors like practitioner and participant reasons for educed risk factor assessments can be incorporated.

Qualitative data like direct questions/interview can be helpful.

Authors’ responses:

Thank you for the helpful comment.

Individual level quantitative patient related factors (such as age, sex, ethnicity, and deprivation) have been included as predictors in the multivariable regression model. Deprivation, as measured by the Index of Multiple Deprivation (IMD), has been considered as a quantitative practice-level factor in our analysis as the IMD is a composite measure that captures the relative deprivation of areas based on multiple domains, including income, employment, education, health, crime, barriers to housing and services, and living environment.

However, unfortunately, we did not have access to practitioner- or participant-related qualitative factors, direct questions, or interviews. Future studies could incorporate these elements to provide a more comprehensive understanding of the underlying reasons and context behind our findings.

Patient Recruitment:

Data set has been taken from a specific population or has been included based on specific criteria needs to be elaborated.

Authors’ responses:

Thank you for the feedback.

The population for this study was drawn from Clinical Practice Research Datalink (CPRD) GOLD, which is a large, representative UK-based database of anonymised electronic health records from a wide network of participating practices across the UK (which indicates a low risk of selection bias).

The cohort selection method was based on a comprehensive set of inclusion and exclusion criteria, as detailed in the Methods section (Data Source and Study Population) and visually represented in Supplementary Figure S2. All individuals meeting the eligibility criteria were included and no sampling method was employed.

Furthermore, we adhered to the Reporting of Studies Conducted Using Observational Routinely Collected Health Data (RECORD) checklist guidelines (reported in the Supplement) during the cohort selection process to ensure transparency, reproducibility, and alignment with best practices for studies using routinely collected healthcare data.

Statistical Analysis:

Author should include multivariate regression and post hoc analysis like in case of gender and ethnic disparities.

Authors’ responses:

Thank you for the suggestion regarding statistical analysis. We have conducted multivariable regression as a secondary analysis and included this in the Methods section:

As a secondary analysis, we used complete case univariable and multivariable logistic regression models to estimate the association of patient characteristics with the outcome variable of a patient having all the selected RFs assessed during a given period (yes vs no). These analyses were performed for the three cohorts and the three time periods separately to explore the temporal trends and differences across the LTCs. Model results are shown as odds ratios (OR) with 95% confidence intervals (95% CI). Age, sex, ethnicity, deprivation, and prevalent comorbidity (T2DM, CVD, CKD, and hypertension) were included as hypothesised predictors.

We have accounted for patient characteristics such as age, sex, and ethnicity as predictors in our multivariable regression models. This approach was informed by previous literature that identified these variables as predictors or confounders associated with the assessments of risk factors, rather than as effect modifiers. Based on this evidence, we did not stratify our models by sex or ethnicity, as our method allows for a more appropriate understanding of the overall trends.

Discussion:

Rephrase the first paragraph of discussion.

Authors’ responses:

Thank you for your suggestion, we have rephrased the first paragraph of the Discussion as below (this can also be viewed in the revised manuscript with tracked change):

Our analysis shows that the assessments of several key risk factors among people with cardiometabolic diseases (T2DM, CVD, or CKD) decreased significantly during the pandemic. This reduction was observed consistently across all three conditions, all RFs examined, and in stratified analyses by age, sex, ethnicity, and deprivation. Additionally, we found an increase in BP levels among individuals with T2DM, leading to a higher proportion of patients with uncontrolled BP during the pandemic. During the pandemic, the likelihood of older patients having all RFs assessed compared to younger patients decreased in all three cohorts. Among those with T2DM, the likelihood of males having all RFs assessed compared to females decreased, whereas in the CKD cohort, males showed a stronger likelihood in 2020.

Discuss about the potential biases in the study. How have you neutralized the confounding factors?

Authors’ responses:

Thank you for highlighting the importance issue. We acknowledge that observational studies like ours are subject to biases.

We took several steps to mitigate biases in our study, including using a large and representative dataset (CPRD GOLD), and conducting multivariable regression models to adjust for potential confounders such as age, sex, ethnicity, and deprivation. However, we acknowledge that biases cannot be entirely eliminated, as residual confounders may remain. Unmeasured factors, such as behavioural influences, healthcare access disparities, or policy changes, could still affect the observed associations and trends. We have included the following limitation in the Discussion section (Strengths and Limitations) as below (this can also be viewed in the revised manuscript with tracked change):

Although we accounted for age, sex, ethnicity, and deprivation in our regression models, we acknowledge that biases cannot be entirely eliminated, as residual confounders may remain. Unmeasured factors, such as behavioural influences, healthcare access disparities, or policy changes, could still affect the observed associations and trends.

How have you handled the missing data or loss to follow up?

Authors’ responses:

We only had missing data for ethnicity and deprivation (IMD), which were minimal. Missing data for ethnicity ranged from 2.1-6.5% across the cohorts, while deprivation data had <0.05% missing values for all the cohorts (Table 1a-c); these proportions remained overall consistent across the study periods. Given the low levels of missingness and their stability over time, we opted not to perform missing data imputation. Instead, we used a complete case analysis approach for our multivariable models.

Loss to follow-up is a common issue in studies with longer follow-up durations. To address this, we divided the study into three distinct time periods within each cohort and assessed risk factor measurements within each one-year time period. All eligible individuals were included in the analysis for the specified timeframe, providing a consistent method to evaluate risk factor assessments. To address the loss to follow up, we have conducted a new sensitivity analysis restricting the cohort to individuals with complete follow-up for each year (added as Supplementary Figure S7). The results of this restricted analysis were consistent with our primary findings, demonstrating the robustness of our conclusions despite variations in follow-up durations.

Include more studies from recent literature and discuss the findings with your study. Authors’ responses:

Thank you for the useful suggestion, we have added two more studies and discussed them in the Discussion section (this can also be viewed in the revised manuscript with tracked change).

Reviewers' comments:

Reviewer #1:

The title requires rewriting.

Authors’ responses:

We have revised as suggested (please see below) and believe it better reflects the content and scope of the study. However, we are happy to consider further suggestions regarding its clarity.

COVID-19 Pandemic and Risk Factor Measurement in Individuals with Cardio-Renal-Metabolic Diseases: A Retrospective Study in the United Kingdom

In the Introduction section, state the risk factors of COVID 19 Disease.

Authors’ responses:

As suggested, we have revised the Introduction section to include the risk factors of COVID-19 (this can also be viewed in the revised manuscript with tracked change):

COVID-19, caused by the SARS-CoV-2 virus, has disproportionately impacted individuals with pre-existing health conditions, with evidence identifying several key risk factors associated with severe disease and mortality. These include older age, male sex, socioeconomic deprivation, ethnicity, obesity, and chronic conditions such as diabetes, cardiovascular, and respiratory diseases. These risk factors are particularly concerning in populations with cardio-renal-metabolic conditions, as these individuals often experience impaired immune responses when infected with SARS-CoV-2 [5].

Reference:

5. Bhaskaran K, Bacon S, Evans SJ, Bates CJ, Rentsch CT, MacKenna B, et al. Factors associated with deaths due to COVID-19 versus other causes: population-based cohort analysis of UK primary care data and linked national death registrations within the OpenSAFELY platform. Lancet Reg Health Eur. 2021;6:100109.

The sampling method should be specified.

Authors’ responses:

The population for this study was drawn from Clinical Practice Research Datalink (CPRD) GOLD.

The cohort selection method was based on a comprehensive set of inclusion and exclusion criteria, as detailed in the Methods section (Data Source and Study Population) and visually represented in Supplementary Figure S2. All individuals meeting the eligibility criteria were included and no sampling method was employed.

The discussion is very complicated and makes it heavier for the reader also needs to become stronger.

Authors’ responses:

We appreciate the reviewer for the valuable comment. We have revised the Discussion section to make it clearer. We focused on key findings and their implications and request the reviewer to read our responses above and below regarding more changes in the Discussion. The revised Discussion can be viewed in the updated manuscript with tracked change.

The conclusion is very weak and requires re-examination and rewriting.

Authors’ responses:

We appreciate the feedback. We have revised the Conclusion to make it clearer and stronger (this can also be viewed in the revised manuscript with tracked change):

In conclusion, our study highlights a substantial reduction in the assessments of several key RFs and poor control in those assessed during the pandemic among people with major long-term conditions. These findings underscore critical gaps in preventative care and chronic disease management during public health emergencies. To address these issues, healthcare policies must prioritise the integration of resilient systems capable of maintaining regular monitoring and management of high-risk groups during acute crises.

Reviewer #2: This retrospective study offers valuable insights by analyzing a wealth of secondary data to compare patients' clinical conditions pre- and peripandemic. While the findings are well-presented, the study could benefit from a clearer application of results to inform future policymaking. Specifically:

Practical Implications:

lEmphasize how findings translate into actionable recommendations for policy and practice, especially in preparation for future scenarios involving resource constraints.

Authors’ responses:

As suggested, we have added this in the Discussion section under the sub-headline Implications for policy (this can also be viewed in the revised manuscript with tracked change):

To address risk factor management disruptions, healthcare policies must prioritise the integration of resilient systems capable of maintaining regular monitoring and management of high-risk groups, even during crises. Specifically, increasing the use of telemedicine, home monitoring tools, and targeted outreach programs would ensure continuity of care and prevent worsening outcomes in vulnerable populations. Additionally, integrating disaster medicine specialty into healthcare planning can improve preparedness and response during emergencies. Our study provides evidence to inform policies aimed at strengthening primary care infrastructure, and implementing proactive measures to miti

---

## [Editor Report · Decision Letter 1]

3 Feb 2025

COVID-19 Pandemic and Risk Factor Measurement in Individuals with Cardio-Renal-Metabolic Diseases: A Retrospective Study in the United Kingdom

PONE-D-24-38823R1

Dear Dr. Khunti,

We’re pleased to inform you that your manuscript has been judged scientifically suitable for publication and will be formally accepted for publication once it meets all outstanding technical requirements.

Kind regards,

Apeksha Niraula, M.D.

Academic Editor

PLOS ONE
---

## [Editor Report · Acceptance letter]

PONE-D-24-38823R1

PLOS ONE

Dear Dr. Khunti,

I'm pleased to inform you that your manuscript has been deemed suitable for publication in PLOS ONE. Congratulations! Your manuscript is now being handed over to our production team.

Kind regards,

on behalf of

Dr. Apeksha Niraula

Academic Editor

PLOS ONE